# HIV-1 Infection Results in Sphingosine-1-Phosphate Receptor 1 Dysregulation in the Human Thymus

**DOI:** 10.3390/ijms241813865

**Published:** 2023-09-08

**Authors:** Rachel S. Resop, Bradley Salvatore, Shawn J. Kim, Brent R. Gordon, Bianca Blom, Dimitrios N. Vatakis, Christel H. Uittenbogaart

**Affiliations:** 1Department of Microbiology, Immunology and Molecular Genetics, University of California, Los Angeles, CA 90095, USA; resopr@ucla.edu (R.S.R.); bradley.salvatore@ucla.edu (B.S.); jyk3093@gmail.com (S.J.K.);; 2UCLA AIDS Institute and Center for AIDS Research, University of California, Los Angeles, CA 90095, USA; vatakisd@gmail.com; 3Amsterdam University Medical Centers, University of Amsterdam, 1105 AZ Amsterdam, The Netherlands; b.blom@amc.uva.nl; 4Department of Medicine, Division of Hematology-Oncology, David Geffen School of Medicine at UCLA, Los Angeles, CA 90095, USA; 5Department of Pediatrics, David Geffen School of Medicine at UCLA, University of California, Los Angeles, CA 90095, USA; 6Jonsson Comprehensive Cancer Center, University of California, Los Angeles, CA 90095, USA

**Keywords:** sphingosine-1-phosphate, sphingosine-1-phosphate receptor, HIV, thymus, humanized mice

## Abstract

Regeneration of functional naïve T lymphocytes following the onset of human immunodeficiency virus (HIV) infection remains a crucial issue for people living with HIV (PLWH), even when adhering to antiretroviral therapy (ART). Thus far, reports on the impact of HIV-1 infection on the entry of thymic precursors and the egress of functional naïve T lymphocytes to and from the thymus are limited. We examined the impact of HIV-1 on Sphingosine-1-phosphate (S1P) signaling, which governs the egress of functional naïve thymocytes from the thymus to the periphery. Using in vitro experiments with primary human thymocytes and in vivo and ex vivo studies with humanized mice, we show that HIV-1 infection results in upregulation of the expression of S1P receptor 1 (S1PR1) in the human thymus. Intriguingly, this upregulation occurs during intrathymic infection (direct infection of the human thymic implant) as well as systemic infection in humanized mice. Moreover, considering the dysregulation of pro- and anti-inflammatory cytokines in infected thymi, the increased expression of S1PR1 in response to in vitro exposure to Interferon-Beta (IFN-β) and Tumor Necrosis Factor-Alpha (TNF-α) indicates that cytokine dysregulation following HIV infection may contribute to upregulation of S1PR1. Finally, an increased presence of CD3hiCD69− (fully mature) as well as CD3hiCD69+ (less mature) T cells in the spleen during HIV infection in humanized mice, combined with earlier expression of S1PR1 during thymocyte development, suggests that upregulation of S1PR1 may translate to increased or accelerated egress from the thymus. The egress of thymocytes that are not functionally mature from the thymus to peripheral blood and lymphoid organs may have implications for the immune function of PLWH.

## 1. Introduction

A significant issue for people living with HIV (PLWH) is the regeneration of functional naïve T lymphocytes following the onset of infection. T cell reconstitution following the acute phase of HIV-1 infection is incomplete in many individuals despite successful antiretroviral therapy [1,2] and is associated with persistent immune activation [3,4,5]. Increased levels of pro-inflammatory cytokines, such as Interleukin (IL)-6, IL-7, IFN-α, and TNF-α; increased chemokines, such as IP-10 (CXCL10); and the expression of the activation markers CD69 and CD38/HLA-DR on T cells during HIV-1 infection are characteristic signs of continuing immune activation and may contribute to mechanisms hindering T cell reconstitution [4,6,7].

There are limited reports on the mechanisms regulating the entry of progenitor cells into the human thymus and the egress of mature T cells to the periphery, and to date, no data exist describing the impact of HIV-1 infection on these processes. Murine studies first showed that sphingosine-1-phosphate (S1P) and one of its G-protein-coupled transmembrane receptors, S1P receptor 1 (S1PR1), play essential roles in the egress of naïve T cells from the thymus to the periphery [8,9] as well as the egress of memory T cells from the secondary lymphoid tissues [10,11]. We described an analogous role for S1PR1 in the human thymus, showing that S1PR1 is the main receptor expressed in the human thymus and is thereby likely responsible for the S1P-mediated exit of mature human T lymphocytes from the thymus to the blood. We determined that the CD3hiCD27+CD69−CD45RA+CD62L+ subset of mature human thymocytes expresses S1PR1 and responds to S1P [12]. Whether HIV infection of the thymus compromises S1PR1 expression and signaling has not been examined to date.

S1P, a lysophospholipid intra/intercellular signaling molecule, plays a myriad of roles in the human body, many of them in immune function. Its functions in inflammation, migration, the apoptosis/proliferation balance, membrane integrity, cellular adhesion, and cytoskeletal rearrangement are modulated by binding of the ligand (S1P) to S1PR1-5, which are expressed to various degrees across different cells and tissues (reviewed in [13]). In this work, we built upon our earlier studies and utilized well-established humanized mouse models [14,15], in conjunction with ex vivo and in vitro assays, to examine the expression and function of S1PR1 in human thymic implants in mice following HIV infection. We examined the expression of S1PR1 mRNA and expression of the receptor’s transcriptional regulator, Krüppel-Like Factor 2 (KLF2), in thymocytes by real-time quantitative RT-PCR and examined thymic S1PR1 by flow cytometry in the context of HIV infection, which we have previously characterized in non-infected thymocytes [12]. Moreover, we investigated the potential involvement of Type I interferons in S1PR1 changes by anti-interferon receptor antibody treatment of HIV-infected humanized mice. We examined the functional responses to S1P as well as the impact of inflammatory cytokines on the expression of S1PR1. Finally, we assessed the potential impacts on T-cell egress in a humanized mouse model.

Our studies show that HIV-1 infection alters S1P receptor expression in the human thymus. Assessment of pAkt signaling by the mature CD3hiCD69− subset following exposure to S1P suggests that S1PR1 on total and mature CD3hiCD69− human thymocytes appears to remain functional during HIV infection. We also observed an increase in mature and less-mature T cells in the spleen of HIV-infected humanized mice. Thus, the increase in S1PR1 protein expression in the less-mature CD3hiCD69+ thymocyte population in the context of infection may result in increased egress from the thymus to the peripheral blood and lymphoid organs of S1PR1+ cells that are likely less functionally mature in HIV-infected individuals than in non-infected individuals.

## 2. Results

### 2.1. Dysregulation of S1P Receptor 1 Expression Is Observed in Human Thymocytes in HIV-Infected Humanized Mice

To determine whether HIV-1 infection causes changes in S1PR1 expression in the human thymus, we utilized Human Immune System (HIS)-Rag2^−/−^γ-chain^−/−^ mice implanted with human fetal thymus/liver (thy/liv) tissue. HIS mice were infected intrathymically (direct injection of virus into the implant) with CXCR4-tropic (NL4-3) or mock (empty vector) HIV-1 and sacrificed at 5 or 9 weeks post-infection. Well-established infection was verified by real-time quantitative reverse transcription PCR (RT-qRT-PCR) for multiply spliced tat/rev mRNA in thymocytes. We performed RT-qRT-PCR for S1PR1 messenger RNA (mRNA) relative to the housekeeping gene GAPDH on total mock and HIV-infected thymocytes at weeks 5 and 9 post-infection (Figure 1A,D; schematics). Further, we examined the expression of the anti-proliferative transcriptional regulator Krüppel-Like Factor 2 (KLF2) [16,17,18] which is involved in the regulation of T cell egress from the thymus [19] and has been shown to regulate the expression of S1PR1, CD62L, CCR3, and CCR5 [20]. There was a trend toward increasing S1PR1 mRNA expression in NL4-3-infected thy/liv implants of HIS mice relative to mock-infected implants at 5 weeks (Figure 1B) and a statistically significant elevation (*p* = 0.0381) in NL4-3-infected mice at 9 weeks (Figure 1E). We also found a trend toward the increasing expression of KLF2 in total thymocytes from NL4-3-infected implants at 5 weeks (Figure 1C) and a significant elevation of KLF2 (*p* = 0.0061) in total thymocytes from NL4-3-infected implants at 9 weeks (Figure 1F).

### 2.2. Interferon-Beta (IFN-β) Increases S1PR1 Expression in Thymocytes

Interferons and interferon-stimulated genes are upregulated in peripheral blood and lymph nodes in treatment-naïve HIV-1 infected individuals [21,22,23,24]. We expanded on the current body of knowledge by first examining the effect of IFN-α2 on S1PR1 expression, exposing postnatal thymocytes in vitro to two concentrations of IFN-α2 for 18 h (1000 and 10,000 U/mL) and examining S1PR1 expression by RT-qRT-PCR. Following 18 h of treatment, S1PR1 mRNA was downregulated on total thymocytes when exposed to the higher concentration of IFN-α. Consequently, we examined whether another Type I interferon, IFN-β, impacts the expression of S1PR1. Postnatal thymocytes were cultured in serum-free media with increasing concentrations (3-300U) of recombinant IFN-β for 24–48 h. S1PR1 expression was then quantified on mature CD3hiCD69− and CD3hiCD69+ thymocyte subsets. As shown in Figure 2A,B, IFN-β increased S1PR1 expression on mature CD3hi thymocytes at 48 h in a dose-dependent manner (n = 8). A one-way ANOVA test was used to assess the effect of IFN-β on S1PR1 expression. Tukey’s HSD Test for multiple comparisons showed that there was a statistically significant increase in S1PR1 expression on mature CD3hiCD69− thymocytes in the presence of 300U of IFN-β (*p* = 0.0078) as well as on the CD3hiCD69+ thymocyte subset (*p* = 0.0078). These results reflect one potential contributing factor to the effect of HIV-1 on S1PR1 expression that we observed in mature thymocyte subsets ex vivo from infected and mock-infected thy/liv implants (presented in Figure 1).

We next examined whether the expression of the IFN receptors (IFNAR1/2) may contribute to the observations reported above. IFN-α and IFN-β utilize the same receptors (IFNAR1/IFNAR2) but have different binding affinities and functions; IFN-β has a higher affinity (50-fold higher) for IFNAR1 than IFN-α2 [25,26]. To determine the thymic expression of IFNAR1/2, primary human postnatal thymocytes were stained with antibodies to IFNAR1 and IFNAR2 in combination with cell surface markers to distinguish immature and mature thymocyte subsets. We found that human thymocytes express lower levels of IFNAR1 than IFNAR2 across all four thymocyte subsets examined. Although the expression of IFNAR2 was higher than that of IFNAR1 in subsets reflecting all stages of thymocyte maturation, IFNAR1 was also detected in all subsets (Appendix A). IFNAR2 expression was statistically significantly greater in the most immature thymocyte subset than IFNAR1 expression (*p* = 0.0159). Thus, our in vitro experiments with exogenous IFN-α and IFN-β suggest that the increase in S1PR1 that we observed on thymocytes during HIV infection may be more related to endogenous IFN-β than IFN-α and may be mediated to a greater extent by IFNAR2 than IFNAR1.

### 2.3. Systemic HIV Infection Results in Increased S1PR1 Expression on Thymocytes That Is Abrogated by Blocking IFNAR2 In Vivo

To consolidate the ex vivo and in vitro findings described above, we used NSG-BLT humanized mice systemically infected with dual-tropic HIV 89.6 for 12 weeks, from which we examined human thymic implants (generously provided by Dr. Anjie Zhen at UCLA, experimental schematic Figure 3A). To elucidate the possible thymocyte populations in which S1PR1 expression was upregulated in our earlier observations, thymocytes were analyzed by flow cytometry for expression of S1PR1 and several additional thymocyte phenotyping markers corresponding to stages of human thymocyte development: CD69, CD27, CD45RA, CD8, CD4, CD3, CD25, and CD62L, plus viability dye, as previously done [12]. We gated on CD3−CD69− and CD3loCD69+ populations within CD27- (cortical) thymocytes as well as CD3hiCD69+ and CD3hiCD69− thymocytes within CD27+ (medullary) thymocytes (gating schematic, Appendix A). We have previously described that the highest S1PR1 expression is observed on CD3hiCD69− human thymocytes in uninfected fetal, postnatal, and adult thymi [12]. Therefore, we hypothesized that the CD3hiCD69− population would be responsible for the observed overall increase in S1PR1 in HIV-infected thy/liv implants.

In these systemically infected animals, we observed upregulation of S1PR1 by flow cytometry in human thy/liv implants in both the CD3hiCD69+ and CD3hiCD69− human thymocyte subsets (Figure 3B–D). A one-way ANOVA was performed to compare the effects of different treatment groups on S1PR1 expression. In contrast to the experiments described above, these animals were infected systemically, not intrathymically, which suggests that the effect on receptor expression likely stems from a systemic, cytokine-mediated effect of HIV-1 rather than only localized effects in the thymus. Interestingly, the increase in S1PR1 was abrogated by the administration of an IFNAR2 blocking antibody (αIFNAR2, [15]), with the greatest reduction in S1PR1 expression observed with dual treatment of ART (tenofovir/emtricitabine/raltegravir) and αIFNAR2 (*p* = 0.0003 for HIV vs. ART + αIFNAR2 in CD3hiCD69+ and *p* < 0.0001 for HIV vs. ART + αIFNAR2 in CD3hiCD69−, Figure 3B–D). This supports a key role for interferons in the modulation of S1PR1 expression during HIV infection and, considering our in vitro data characterizing IFNAR2 in the human thymus, supports a potential role for IFN-β in increasing S1PR1 expression via IFNAR2. Further studies will be needed to understand the roles of IFN-α, IFN-β, and IFNAR1/2 in modulating the expression of S1PR1 in the context of HIV-1 infection.

Unexpectedly, we noted upregulation of S1PR1 in a population of less mature thymocytes that normally have limited expression of the receptor, the CD3hiCD69+ population (Figure 3B,C). Thy/liv implants from mock-infected animals expressed minimal levels of S1PR1 within this population. Implants from HIV-infected animals, when compared to implants from mock mice, had notably higher levels of S1PR1+ cells within the CD27+CD3hiCD69+ population (Figure 3C). The expression of S1PR1 on a less mature CD3hiCD69+ thymocyte subset that is normally not prepared to egress the thymus suggests that HIV infection may induce more rapid T cell differentiation, as has been observed in acute SIV infection in Rhesus Macaques [27]. Therefore, this is of potential importance for understanding T cell regeneration in HIV infection. Thus, expression of S1PR1 is enhanced during acute HIV infection; moreover, it is enriched in populations in which the receptor is normally not expressed, suggesting that immature thymocytes may have the ability to emigrate from the thymus before maturation is complete.

### 2.4. Tumor Necrosis Factor Alpha (TNF-α) Likely Contributes to S1PR1 Upregulation in HIV Infection along with IFN-β

We examined HIV-infected and mock-infected thy/liv implants by RT-qRT-PCR to determine whether additional cytokines were perturbed during persistent infection of the thymus, as reports of cytokine dysregulation in the thymus during HIV-1 vary by model system [27,28,29,30,31]. We found that in addition to IFN-α, tumor necrosis factor alpha (TNF-α) mRNA was elevated by 5 weeks in NL4-3-infected thy/liv implants and remained elevated at the 9-week time point (*p* = 0.023, Appendix A). In addition to IFN-α, IFN-β, and TNF-α are cytokines that are constitutively produced in the thymus and participate in the well-documented cytokine storm during HIV infection [32]. As described above, IFN-α2, one of the IFN-α subtypes, did not upregulate S1PR1, but IFN-β did upregulate S1PR1 in our model. To determine whether TNF-α may also influence S1PR1, we exposed human postnatal thymocytes to TNF-α in vitro for 18 h and examined S1PR1 mRNA by RT-qRT-PCR. We found that following the 18 h incubation with TNF-α, S1PR1 mRNA was upregulated relative to the untreated donor-matched control with three concentrations of TNF-α: 50, 100, and 500 ng/mL, with the greatest effect observed at 100 ng/mL. As TNF-α appeared to influence S1PR1 expression at the mRNA level, we examined S1PR1 protein expression by flow cytometry following 24 h of exposure to exogenous TNF-α at the same three concentrations. TNF-α-treated human postnatal thymocytes (CD27+CD3hiCD69−) demonstrated a statistically significant fold increase in S1PR1 mean fluorescence intensity (MFI) at 100 and 500 ng TNF-α, but not at 50 ng TNF-α (n = 7, *p* = 0.0312 for 100 ng and *p* = 0.0156 for 500 ng). Interestingly, S1PR1 percent expression was not consistently increased within this subset; however, the MFI of the mature S1PR1-expressing subset was consistently increased (Appendix A).

As discussed, we observed that exogenous IFN-β resulted in upregulation of S1PR1 protein on thymocytes (Figure 2A,B). The greatest upregulation was observed at 300U of exogenous IFN-β and was within the CD3hiCD69− mature thymocyte population, as well as the CD3hiCD69+, slightly less mature thymocyte population, as we noted in HIV-infected thy/liv implants. Taken together, these results indicate that IFN-β and TNF-α likely contribute to the heightened S1PR1 expression we observed in infected thy/liv implants. A potential mechanism is that cytokines other than IFN-α that are elevated in HIV-infected thy/liv implants may override the requirement for downregulation of CD69 prior to S1PR1 expression, considering our observation that CD27+CD3hi thymocytes in infected thy/liv implants had increased S1PR1 while CD69 was still expressed (CD27+CD3hiCD69+; Figure 3B,C), while only a small percentage of this population in mock-infected thy/liv implants expressed S1PR1.

### 2.5. Neither Total Nor Mature CD3hiCD69− Thymocytes from Infected Thy/Liv Implants Have Impaired Akt Signaling in Response to S1P Ex Vivo Stimulation

We further examined whether populations in the human thymus that upregulate S1PR1 during HIV infection respond to S1P. We assessed phosphorylated Akt (pAkt) in total and mature (CD3hiCD69−) thymocyte populations following S1P exposure, as pAkt has been previously utilized as a proxy measurement for S1PR1 downstream signaling [33]. Total thymocytes prepared from human fetal thy/liv implants from NSG-BLT mice infected with dual-tropic HIV-1 (89.6) for two weeks were exposed to the approximate physiological concentration of S1P, 100 nM [34,35,36], or serum-free medium alone for 30 min. Following S1P exposure, thymocytes were immediately placed on ice and stained with human thymocyte surface markers. Cells were then briefly fixed with 1.6% paraformaldehyde prior to permeabilization with cold methanol and subsequent staining with anti-phosphorylated Akt (pAkt). We gated on mature thymocytes using surface expression of CD3 and CD69 to examine pAkt signaling by mature thymocytes (the population that expresses S1PR1). We attempted to stain for S1PR1 as well in separate experiments, but as this did not produce reliable, distinct populations once cells were permeabilized, we focused on mature CD3hi69− thymocytes, which comprise the main population of S1PR1-expressing human thymocytes.

We determined the fold change in pAkt following S1P exposure in total thymocytes as well as in mature CD3hiCD69− thymocytes. Following S1P exposure, there was no significant difference in fold change of pAkt between HIV-infected vs. uninfected total or mature thymocytes (Figure 4A,B). This suggests that the population of thymocytes that upregulates S1PR1 likely retains Akt signaling activity and that these thymocytes are functionally responsive to S1P during HIV infection.

### 2.6. HIV Infection Increases the Egress of Thymocytes to the Spleen in Humanized Bone Marrow Thymus/Liver Triple Knockout (BLT-TKO) Mice

To determine whether HIV infection affects the egress of S1PR1+ thymocytes to the periphery, we used the triple knock-out (TKO) HIS mouse model, the C57BL/6 Rag2^−/−^CD47^−/−^ILRG^−/−^ line [37]. The TKO bone marrow/liver/thymus BLT humanized mice develop peripheral lymphoid tissues with human cell subsets and human immunohistochemistry. These mice can be infected with CCR5-tropic HIV by intraperitoneal and mucosal routes and develop T and B-cell immune responses to HIV [37]. They were therefore useful in addressing our question of the effect of CCR5-tropic HIV-1 infection on the egress of thymocytes to peripheral lymphoid organs while mimicking a physiological setting.

Egress of thymocytes to the periphery in mock and HIV-infected TKO mice was measured via flow cytometry by gating on human CD45+ cells in the spleens of mice at 10 weeks post-infection (Figure 5A). At this point, there was no statistically significant change in S1PR1 levels or the proportion of mature human CD3hiCD69+ or CD3hiCD69− T cells in the thymus in TKO mice infected with CCR5-tropic HIV as compared to mock-infected TKO mice (*p* = 0.8571 and *p* > 0.9999, CD3/69 populations, Figure 5B,C). However, there was a statistically significant increase in the presence of mature human CD3hiCD69+ cells as well as CD3hiCD69− T cells (*p* = 0.0357 and *p* = 0.0357, Figure 5D,E) in the spleen of HIV-infected mice as compared to mock-infected mice. No significant changes in other human thymocyte subsets in the spleen of HIV-infected mice were observed. These results demonstrate that HIV infection results in an increased number of mature and less mature human T cells in the periphery, likely as a result of elevated egress from the thymus, as was reported in acute SIV infection in Rhesus macaques [27]. These results strengthen our observations of dysregulation of S1PR1 expression and signaling in thymic egress during acute HIV infection.

## 3. Discussion

The identification of factors that promote the egress of mature single-positive thymocytes to the periphery is crucial to the study of T cell reconstitution during HIV infection. A transient increase in thymic output following acute HIV infection [38] and SIV infection has been reported [27], as well as an increase in T cell differentiation, along with a lack of proliferation, which was observed in acute SIV infection [27]. Whether an increase in thymic output is maintained for a significant period is unknown, although data from adolescents who were perinatally infected suggest a continuing increase in thymic output [39]. Further, the receptors and chemokines responsible for changes in egress during HIV infection are understudied. We examined the expression and function of S1P receptor 1 (S1PR1), which have thus far not been explored in the context of T cell reconstitution during HIV infection. To perform these studies, we employed various humanized mouse models. Importantly, these advanced in vivo models involve essentially complete humanization of the immune system with primary human cells that reside in several anatomic locations and thus closely mimic the physiological environment of HIV-infected humans. We observed an increase in expression of the chemotactic signaling receptor S1PR1 in HIV-infected thymic implants in humanized mice at 5 and 9 weeks post-infection, as well as in the thymus of systemically infected humanized mice at 12 weeks, which raises the critical question of whether upregulated S1PR1 translates to enhanced egress of functional thymocytes or, alternatively, a detrimental egress of less-functional naïve T cells due to more rapid intrathymic differentiation, as reported in studies of SIV-infected Rhesus macaques [27]. Increased egress of mature thymocytes to the periphery during HIV infection was confirmed in the BLT-TKO humanized mouse model where we observed an increase in human T cells in the spleen of both CD3hiCD69− (fully mature) and CD3hiCD69+ (less mature) T cells 10 weeks post-infection with CCR5-tropic HIV. The increased egress of mature thymocytes to the periphery was not reflected in the thymus, likely due to the relatively low percentage of mature thymocytes within the total population; it is also possible that we may have observed altered S1PR1 expression on thymocyte populations at a different time point. These findings should be further explored to potentially inform the development of novel therapies for HIV patients who do not recover optimal CD4 levels with antiretroviral therapy.

We have previously shown by flow cytometry and RT-qRT-PCR that S1PR1, one of the five S1P receptors, is prevalent in the human thymus and is expressed on mature CD3hiCD 69−CD27+CD45RA+CD62L+ thymocytes that are preparing to egress the thymus to the periphery [12]. Our data show that S1PR1 is upregulated in human thymocytes during intrathymic HIV-1 infection at two time points post-infection as well as in mature thymocyte populations during systemic HIV-1 infection. Interestingly, within a less mature population of thymocytes, the CD3hiCD69+ thymocytes, S1PR1 was also increased in HIV-infected thymi relative to mock-infected thymi. This implies that mechanisms that control S1PR1 expression may be modified during HIV infection, permitting S1PR1 to be expressed at an earlier stage of development than is observed in the non-infected thymus. As mentioned, a transient increase in thymic output during SIV infection of Rhesus macaques, characterized by reduced intrathymic proliferation and a change in the cytokine profile of the thymus allowing for more rapid progression through the thymus microenvironment and egress to the periphery, has been reported [27]. Our observation that S1PR1 is increased during HIV infection of the human thymus and expressed at an earlier stage of development than is normally observed may support the possibility of enhanced thymic output resulting from speedier development in the thymus. Moreover, we observed that KLF2, an anti-proliferative transcription factor [16,17,18] that is also necessary for the regulation of T cell egress from the thymus [19] and has been shown to regulate the expression of S1PR1 as well as other receptors such as CD62L, CCR3, and CCR5 [20], was also upregulated in the HIV-infected thymus at both 5 and 9 weeks post-infection. As KLF2 dampens proliferation and upregulates S1PR1, it is a strong candidate for influencing the changes we observed in S1PR1 expression in HIV infection as well as the changes observed by others in intrathymic proliferation time [38]. We are currently exploring the possibility that KLF2 may be directly activated by HIV accessory proteins and that this mechanism may contribute to elevated S1PR1 in the infected thymus.

Others have shown in murine models that IFN-α, induced following immune activation or viral infection, interferes with the egress of mature thymocytes [11]. In HIV-induced immune activation, increased levels of IFN-α and CD69 are present [6], which are likely to influence S1PR1 receptor expression and thereby the entry of hematopoietic stem cells into the thymus and the exit of naïve T cell subsets to the periphery. Although we hypothesized that interferon elevation during HIV-1 infection would maintain CD69 expression and low levels of S1PR1, we observed, to our surprise, that both S1PR1 mRNA and protein were significantly elevated and not decreased during HIV infection, expanding upon the current understanding of negative regulation of S1PR1 by CD69. Therefore, there appears to be a mechanism that overrides the requirement for the downregulation of CD69 for S1PR1 to be expressed in our models of HIV-1 infection of the human thymus as well as systemic infection. Our observation of S1PR1 upregulation in the less mature CD3hiCD69+ thymocytes corroborates this possibility. We are currently examining the role of KLF2 in superseding the requirement for CD69 downregulation prior to S1PR1 expression.

We tested additional cytokines that are constitutively expressed in the thymus for their ability to change the expression of S1PR1 in vitro. We hypothesized that certain cytokines may contribute to heightened S1PR1 expression, potentially overcoming the effects of IFN-α. Indeed, after 24 h of incubation with IFN-β (Figure 2) or TNF-α (Appendix A), S1PR1 protein was increased on mature CD27+CD3hiCD69− thymocytes. Peak upregulation occurred at 300U IFN-β, whereas peak upregulation with TNF-α occurred at 100 ng/mL, with the effect dampened at higher concentrations for both cytokines. These effects were supported by RT-qRT-PCR, where we observed that 18 h of TNF-α incubation increased S1PR1 mRNA expression. Together, these results indicate that both IFN-β and, to a lesser extent, TNF-α, may contribute to increased S1PR1 expression in the HIV-infected thymus, whereas IFN-α may act to control the receptor’s expression in non-infected thymic tissue. As additional cytokines, some of which are constitutively expressed in the thymus [40,41], may be perturbed by HIV infection, we are unable to definitively assign roles to these cytokines alone in effecting changes in S1PR1 on mature thymocytes. The majority of reports on the “cytokine storm” during immune activation secondary to HIV infection have focused on the periphery [7,32,42,43,44]; therefore, additional work in humanized mouse models to characterize the changes in cytokine profiles in the HIV-infected thymus and the effects of these changes on chemotactic receptor expression and function is needed. Interestingly, there has been a dearth of studies examining S1PR1 expression and inflammation in other viral infections. We identified one study where elevated expression of S1PR1 correlated with increased inflammatory cytokine levels in Newcastle disease virus infection [45].

Our results contribute to the field of T cell reconstitution in an entirely novel manner. S1PR1 is required for T cell egress from the thymus; however, to our knowledge, we are the first to characterize the receptor’s expression and function in the human thymus [12] and investigate alterations in its expression in the thymus in the context of HIV infection. We observe that HIV-1 infection alters S1PR1 expression independent of the humanized mouse model, HIV molecular clone/tropism, or route of infection, and that S1PR1, while elevated, likely remains functional during HIV infection of the thymus as per proxy measurement of pAkt signaling upon exposure to S1P. In addition, we found that the cytokines IFN-β and TNF-α likely contribute to the heightened S1PR1 expression we observed in infected thy/liv implants. Pro- or anti-inflammatory cytokines other than IFN-α that are dysregulated in HIV-infected thy/liv implants [27,28,29,30,31] may override the requirement for downregulation of CD69 prior to S1PR1 expression, in light of our observation that CD27+CD3hiCD69+ thymocytes in infected thy/liv implants upregulate S1PR1 while CD69 remains expressed (Figure 1 and Figure 2), while only a small percentage of this population in mock-infected thy/liv implants expresses S1PR1.

As our data indicate that S1PR1 is upregulated in the context of HIV infection and is functional on not yet fully mature CD3+CD69+ thymocytes, approaches to modulate S1PR1 expression may present an intriguing new possibility for the treatment of patients who lack immune system reconstitution in response to ART.

## 4. Materials and Methods

### 4.1. Tissue Collection and Primary Thymocyte Preparation

Normal human postnatal thymus specimens were obtained from children undergoing corrective cardiac surgery at the UCLA Mattel Children’s Hospital. Thymocytes were prepared and cultured as previously described [12,40,46]. Briefly, tissues were placed in NH_4_Cl-Tris lysing buffer to remove red blood cells, while the tissue was cut into small pieces and passed over a cell strainer to generate a single-cell suspension of thymocytes. Cells were washed in serum-free medium consisting of IMDM (Omega Scientific, Tarzana, CA, USA) supplemented with 1.1 mg/mL delipidated Bovine serum albumin (dBSA) (Sigma-Aldrich, St. Louis, MO, USA), 85 µg/mL human transferrin (Sigma-Aldrich), 2 mM l-glutamine, and 25 U/25 µg/mL penicillin/streptomycin, then resuspended at 4 × 10^7^ cells/mL in serum-free medium.

### 4.2. In Vitro Thymocyte Cultures

Thymocyte cultures were maintained as previously described [12,46]. Briefly, thymocytes were cultured at 2 × 10^7^ cells/mL in serum-free medium supplemented with dBSA, transferrin, and l-glutamine as described above and maintained as pellet cultures at 37 °C in 5% CO_2_ in round-bottom tissue culture tubes. For cytokine experiments, thymocytes were cultured for 48 h with cytokines including IFN-α, IFN-β, and TNF-α at various concentrations prior to collecting cells for flow cytometric analysis or real-time quantitative RT PCR (RT-qRT-PCR). Whenever possible, in vitro assays were initiated on the same day as the thymocyte suspension was prepared.

### 4.3. HIV Molecular Clones

HIV-1 NL4-3 (CXCR4-tropic), Bal (CCR5-tropic), and 89.6 (dual-tropic) molecular clones were used for infection of humanized mice. All molecular clones were obtained from our collaborators at the UCLA AIDS Institute.

### 4.4. Generation of Immunodeficient (HIS)-Rag2^−/−^γ-Chain^−/−^ Mice and Intrathymic Infection with HIV

Human Immune System (HIS)-Rag2^−/−^γ-chain^−/−^ mice were implanted under the kidney capsule with human fetal thymus/liver (thy/liv) tissue, which donated human CD34+ hematopoietic stem cells and human thymocytes, a procedure comparable to the subcutaneous implant protocol we reported previously [14]. Approximately 15 weeks post-transplantation, the animals were injected intrathymically with approx. 300–400 infectious units (IU) of CXCR4-tropic (NL4-3) or mock (empty vector) virus. Animals were sacrificed at 5 or 9 weeks post-infection, and thymi and spleens were obtained for ex-vivo assays, including flow cytometric and RT-qRT-PCR analyses.

### 4.5. Systemic Infection of Immunodeficient Mice with HIV

For our studies of the effect of systemic HIV-1 infection on pAkt signaling and S1PR1 expression (including ART and IFNAR2 blocking studies), we collected human thymic tissues from two different humanized mouse models. Bone marrow/liver/thymus (NSG-BLT) mice were generated as previously described [47]. Four to six weeks after transplantation, the animals were injected retro-orbitally or intraperitoneally with 300–400 infectious units (IU) of either dual-tropic HIV-89.6 or an empty vector (control). Human thymic tissues were harvested 2 or 14 weeks after infection to assess the impact of systemic HIV infection on pAkt signaling and S1PR1 expression, respectively.

In addition, we utilized C57BL/6 Rag2^−/−^γ-chain^−/−^CD47^−/−^triple knockout (TKO) immunodeficient mice as recipients of human fetal liver and thymus (thy/liv) tissue grafts (BLT-TKO mice). Human fetal tissue was purchased from the UCLA/CFAR Gene Therapy Core. Due to the genetic immunodeficiency defect in these mice, they do not reject xenografts, which allowed prolonged studies of immune reconstitution without the issue of Graft vs. Host Disease (GVHD) [37]. Thy/liv tissue was implanted under the kidney capsule and the mice received a retro-orbital injection of 0.5 × 10^6^ additional CD34+ cells obtained from the human fetal liver as well as 0.5 × 10^6^ autologous spleen cells from the TKO mice and 1 × 10^6^ bone marrow cells as per Lavender et al. [37]. Following reconstitution, CCR5-tropic HIV Bal was used to infect mice intraperitoneally. Mock-infected animals were injected with an equal volume of phosphate-buffered saline (PBS).

### 4.6. Antiretroviral Treatment and IFNAR2 Blockade in NSG-BLT Mice

To further assess the impact of HIV infection on S1PR1 expression, HIV-infected NSG-BLT mice (see above) were treated with antiretroviral therapy (ART) and/or an IFNAR2-blocking antibody. Briefly, HIV-infected NSG-BLT mice were treated daily by intraperitoneal injection of 500 μL of tenofovir (8.75 mg/kg)/emtricitabine (13 mg/kg)/raltegravir (17.5 mg/kg) dissolved in PBS. For IFNAR2 blockade experiments, HIV-infected NSG-BLT mice, either untreated or treated with ART, were treated with 100 μg anti-IFNAR2 blocking antibody (clone MMHAR-2) or an IgG isotype control once every 2 days intraperitoneally for 1 week. NSG-BLT mice that were treated with ART received the blocking antibody during the last week of ART treatment [15].

### 4.7. Phosphorylated Akt Intracellular Flow Cytometry Assay and S1P Exposure

Human thymocytes obtained from BLT-NSG mice (infected or mock, a kind gift of the laboratory of Dr. Dimitrios Vatakis at UCLA; see information on mice above and [47]) were stimulated in suspension with 100 nM S1P (Cayman Chemical, Ann Arbor, MI, USA) for 30 min (or suspended in media alone) at 37 °C in serum-free AT + dBSA medium, then immediately placed on ice. Cells were stained on ice for surface markers (CD69 FITC, CD27 PE, CD45RA PerCP-Cy5.5, CD4 PE-Cy7, CD8 APC-eFluor 780, CD3 eFluor 650^NC^, CD25 eFluor 450, and CD62L eFluor 605^NC^). Cells were then briefly fixed in 1.6% paraformaldehyde on ice (10 min), permeabilized with methanol (4C, 10 min), and subsequently stained for intracellular phosphorylated Akt (pAkt) with α-pAkt APC antibody (eBiosciences, San Diego, CA; isotype control also from eBiosciences).

### 4.8. Flow Cytometry

Surface immunophenotyping of thymocytes with unconjugated and directly conjugated antibodies was performed as previously described [15,48]. Monoclonal antibodies (mAb) were procured from R&D (Minneapolis, MN, USA) (unconjugated anti-human S1PR1 (clone 218713, product #MAB2016), unconjugated IgG2b control), eBioscience (PE-conjugated anti-mouse IgG), eBioscience (CD45RA PerCP-Cy5.5, CD8 APC-eFluor 780, CD25 eFluor 450, CD62L eFluor 605^NC^, CD3 eFluor 650^NC^), and Becton Dickinson (CD69 FITC, CD27 APC, and CD4 PE-Cy7). For S1PR1 detection, cells were first stained with an unconjugated anti-human S1PR1 (or IgG2b isotype control), followed by a phycoerythrin-conjugated anti-mouse IgG antibody. Flow cytometry data were acquired on an LSRII or LSRII-HT analyzer (Becton Dickinson) and analyzed with FCS Express (Version 6, De Novo software). Gates were set based on isotype controls, unstained thymocytes, or fluorescence minus one (FMO) controls, as appropriate.

### 4.9. Real-Time Quantitative Reverse-Transcription PCR (RT-qRT-PCR)

RNA was isolated from cells suspended in Trizol Reagent as per the Trizol protocol (ThermoFisher, Waltham, MA, USA). RNA was quantified by Nanodrop, and real-time quantitative reverse transcription PCR (RT-qRT-PCR) was performed to determine the expression of S1PR1 and KLF2 genes relative to the GAPDH internal control. Jurkat cells were used as positive controls for the S1PR1 (S1P receptor) gene and the subsequent generation of standard curves. Pre-designed primer-probe conjugates were obtained from Invitrogen (Life Technologies/Fischer Scientific, Grand Island, NY, USA) for S1PR1 (Assay ID: Hs01922614_s1), KLF2 (Assay ID: Hs00360439_g1), IFN-α1 (Hs00256882_s1), IFN-α2 (Hs00265051_s1), IFN-β (Hs01077958_m1), MxA (Hs00182073_m1), and ISG15 (Hs00192713_m1). Target gene expression was normalized to primers amplifying GAPDH. The data were analyzed in Microsoft Excel, and statistics were performed in GraphPad Prism (Version 9.0) or SAS Studio (see below).

### 4.10. Statistical Analysis

All analyses were conducted with SAS Studio or GraphPad Prism (Version 9.0, GraphPad Software). The data were expressed as means with a standard error of the mean (SEM). The statistical tests used are described in the results and figure legends. Given the small sample size (<50) a Shapiro-Wilk test was used to assess normality. Unless specified in the text, all data passed the normality test, and parametric tests were used. A *p* value inferior to or equal to 0.05 was considered statistically significant.

### 4.11. Ethics Statement

Normal human postnatal thymus specimens were obtained from children undergoing corrective cardiac surgery at the UCLA Mattel Children’s Hospital. Human fetal tissue was purchased from the UCLA/CFAR Gene Therapy Core and was obtained without identifying information. All experiments using human tissues were approved by the UCLA Office of the Human Research Protection Program and the UCLA Institutional Review Board (IRB), which determined that the de-identified human tissues used in the reported experiments do not meet the definition of human subject research.

Animal research carried out in this manuscript was performed under the written approval of the UCLA Animal Research Committee (ARC) in accordance with all federal, state, and local guidelines. Specifically, the experiments were performed strictly according to the guidelines in The Guide for the Care and Use of Laboratory Animals of the National Institutes of Health and the accreditation and guidelines of the Association for the Assessment and Accreditation of Laboratory Animal Care (AALAC) International under UCLA ARC Protocol Number 1994-232. Veterinary care was provided by the UCLA Vivarium, the Division of Laboratory Medicine, and the Humanized Mouse Core Staff. Animals are observed daily, and core personnel administer medications as directed by the veterinarian. Animals were sacrificed in accordance with the American Veterinary Medical Association Guidelines for the Euthanasia of Animals. All experiments align in accordance with the ARRIVE guidelines. All experiments and agents used were carried out and used in accordance with the UCLA Institutional Biosafety Committee (IBC).

### 4.12. Data Availability

The datasets used and/or analyzed during the current study are available from the corresponding author upon reasonable request.

## Figures and Tables

**Figure 1 ijms-24-13865-f001:**
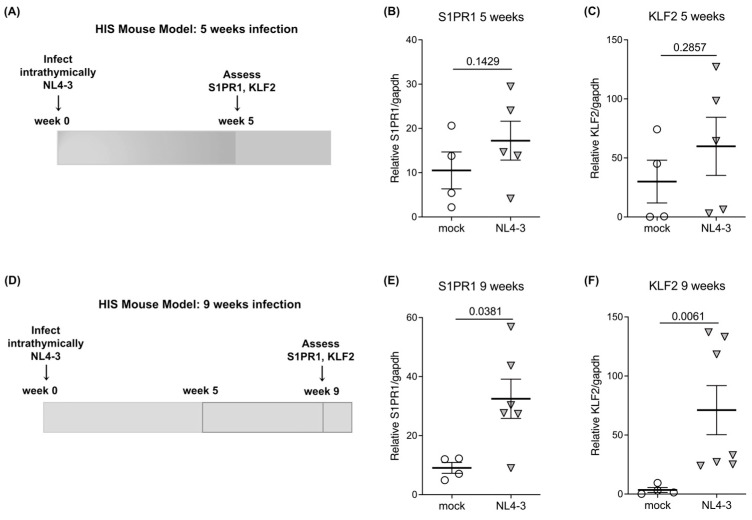
S1PR1 and KLF2 are increased during HIV infection in the thymus at 5 and 9 weeks. (**A**) Schematic of HIS mouse model including infection with NL4-3 (n = 5) or mock-infected (n = 4) and ex-vivo assays performed at 5 weeks post-infection. (**B**) and (**C**) S1PR1 and KLF-2 mRNA expression increase at 5 weeks post-infection. The expression of each gene relative to the housekeeping gene (GAPDH) expression is shown. (**D**) Schematic of the HIS mouse model including infection with NL4-3 (n = 7) or mock-infected (n = 4) and ex-vivo assays performed at 9 weeks post-infection. (**E**) and (**F**) S1PR1 and KLF2 mRNA expression increase at 9 weeks post-infection. The expression of each gene relative to the housekeeping gene (GAPDH) expression is shown. In Figure (**B**,**C**) and (**E**,**F**), the mean with the standard error of the mean (SEM) is shown and two-tailed unpaired t-tests were used for all comparisons.

**Figure 2 ijms-24-13865-f002:**
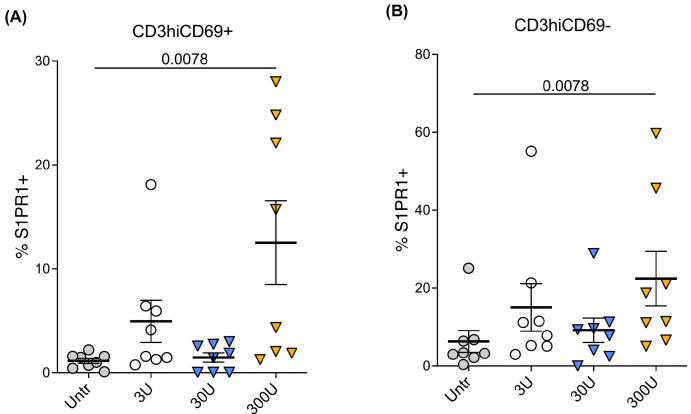
Exogenous IFN-β increases S1PR1 protein expression in CD3hiCD69+ and CD3hiCD69− thymocytes in vitro. Thymocytes were cultured with different concentrations of IFN-β or media alone for 48 h (n = 8 individual thymus donors). S1PR1 expression in thymocyte subsets was measured by flow cytometry in eight different thymus tissues. (**A**) S1PR1 expression in CD3hiCD69+ thymocytes with different concentrations of IFN-β or media alone; (**B**) S1PR1 expression in mature CD3hiCD69− thymocytes with different concentrations of IFN-β or media alone. The mean with the standard error of the mean (SEM) is shown. Data were analyzed by one-way ANOVA using a Tukey HSD test for multiple comparisons. Overall, the data are statistically significant with *p* < 0.001. Tukey’s HSD Test for multiple comparisons found that the mean value of S1PR1 expression was significantly different between 300U and untreated (*p* = 0.0078 for both thymocyte populations).

**Figure 3 ijms-24-13865-f003:**
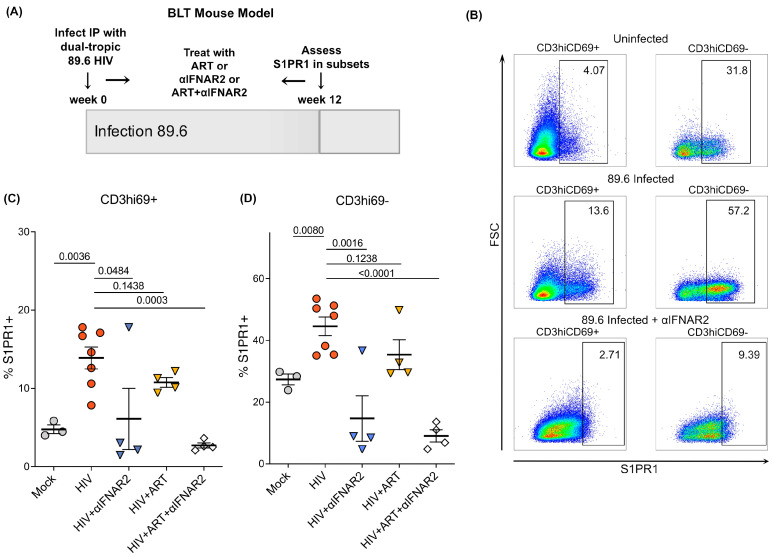
S1PR1 protein expression is increased in BLT mice systemically infected with dual-tropic HIV-1 and is abrogated by IFNAR2 blockade. (**A**) Schematic of infection of BLT mice with dual-tropic HIV (89.6) and analysis of thymic implants (n = 3–7 depending on treatment; n = 22 total). (**B**) S1PR1 expression is increased in two CD3/69 subsets in BLT mice infected with dual-tropic HIV-1 for 12 weeks. Expression was quantified by flow cytometry on CD3hiCD69+ and CD3hiCD69− thymocytes. Shown are three representative animals (mock, HIV and HIV + αIFNAR2) of a total n = 3 mock, n = 7 HIV-infected, n = 4 HIV + αIFNAR2-treated, n = 4 HIV + ART-treated, and n = 4 HIV + ART + αIFNAR2-treated mice. Gates for each condition were set based on isotype controls. (**C**,**D**) Summary of S1PR1 expression in CD3/69 thymic subsets in BLT mice infected with dual-tropic HIV-1 at 12 weeks post-infection. The mean with the standard error of the mean (SEM) is shown. Data were analyzed by one-way ANOVA using a Tukey HSD test for multiple comparisons. Overall, the data are statistically significant with *p* < 0.001.

**Figure 4 ijms-24-13865-f004:**
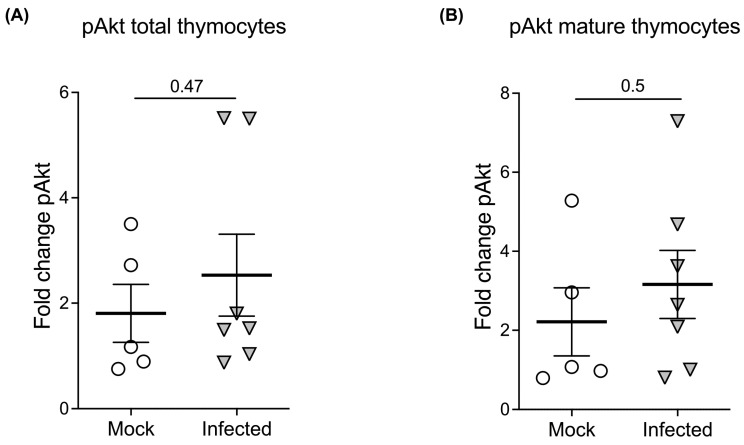
Akt signaling is not impaired during HIV infection in the thymus. Akt phosphorylation was examined by phospho-flow staining following two weeks of infection with dual-tropic HIV in BLT mice (n = 5 mock and n = 6 HIV-1; n = 11 total). (**A**) There is no significant difference in the percentage of pAkt-expressing total thymocytes between mock and HIV-infected animals (*p* = 0.47 by unpaired two-tailed t-test) upon exposure of thymocytes to S1P for 30 min. (**B**) There is no significant difference (*p* = 0.5) in the percentage of pAkt-expressing mature CD3hiCD69− thymocytes between mock and HIV-infected animals by two-tailed unpaired t-test.

**Figure 5 ijms-24-13865-f005:**
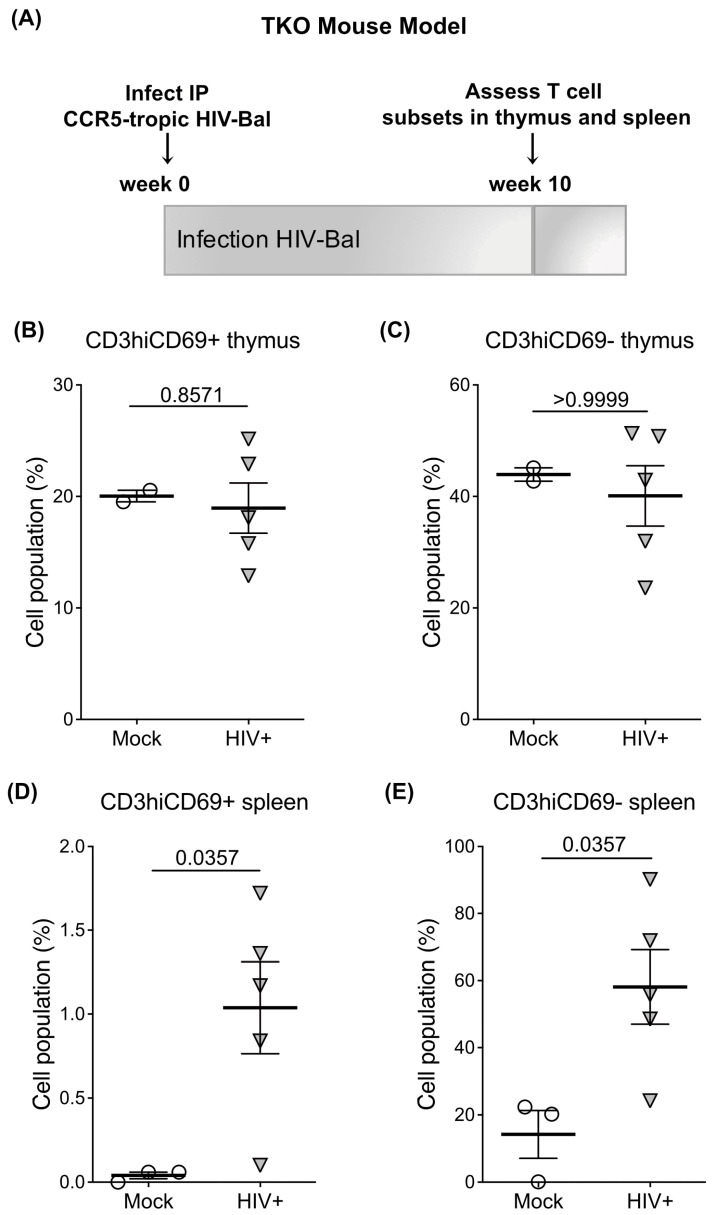
Dynamics of lymphocyte egress in HIV-infected BLT-TKO mice. (**A**) Schematic of TKO mouse model of infection with CCR5-tropic HIV (Bal) and analysis of thymic and splenic human CD45+ T cell subsets at 10 weeks post-infection (n = 3 mock and n = 5 HIV-1; n = 8 mice total). (**B**) CD3hiCD69+ T cells in the thymus of Bal (n = 5) and mock (n = 3) infected BLT TKO mice. (**C**) CD3hiCD69− T cells in the thymus of Bal and mock-infected BLT-TKO mice. (**D**) CD3hiCD69+ T cells in the spleen of Bal and mock-infected BLT TKO mice. (**E**) CD3hiCD69− T cells in the spleen of Bal and mock-infected BLT-TKO mice. The mean with the standard error of the mean (SEM) is shown. All comparisons were done by unpaired t-test.

## Data Availability

The datasets used and/or analyzed during the current study are available from the corresponding author on reasonable request.

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
