# Peer review of "HIV-1 Infection Results in Sphingosine-1-Phosphate Receptor 1 Dysregulation in the Human Thymus"

_ijms, 2023, doi:10.3390/ijms241813865_

Round 1
Reviewer 1 Report
Resop et al. reported S1PR1 dysregulation during HIV-1 infection in human thymus using several humanized mouse models and in vitro cell culture experiments. They found S1PR1 expression is increased during HIV infection regardless of ART treatment, however, this effect can be inhibited by anti-IFNAR2. In vitro treatment of TNF-a and IFN-b can also increase S1PR1 expression in thymocytes. They also found that akt signaling is not impaired in the thymus during HIV infection. Since S1PR1 is upregulated in both CD69 positive and negative cells, the effect of CD69 on S1PR1 expression will need further investigation. Authors suggested that S1PR1 modulation may present a new therapeutic strategy for patients who lack immune system reconstitution due to the increased egress of immature thymocytes to the periphery.
Comments:
1. Line 65: I am not an immunologist but my understanding of “mature” thymocytes means that they have become CD4 or CD8 single positive T cells. CD25 and CD69 are normally used as activation markers. Please explain why only CD8 is included in the flow panel, and any no CD8 nor CD4 results were shown to indicate the actual maturation of the thymocytes, especially in the humanized models when thymus epithelium cells are present.
2. Viral input (TCID) used for infecting mice should be included in the methods. Authors should also explain why 5 weeks, 9 weeks, and 12 weeks post infection were chosen since most of the background information is based on the findings in the acute infection phase.
3. Line 102: The reason for examining KLF2 should be explained in the introduction or result 2.1, not in the discussion.
4. Figure 1 (and most of the other data): authors should show how many percentages of these thymocytes were actually infected by HIV using flow. Flow cytometry gating strategies should include HIV p24 or other viral antigen to detect which cells were actually infected.
5. Figure 1 shows very high real time RT-PCR results presented as relative ratio compared to GAPDH. Some data points are close to 150-fold to GADPH. These are quite unusual real-time results. Or does GAPDH particularly low in the thymocytes?
6. Line 121 and 122: Is the concentration of IFN-a2 1000U or 10000U in mL? Are they physiologically relevant concentrations? Why isn’t the data included in the manuscript? Same to Line 126, is it U/ml?
7. Result 2.5: Why isn’t the phosphor-akt level increased by the increased of S1PR1? Therefore, there is no indication that the S1PR1 signaling is up-regulated by the increased expression of S1PR1. Cells should be gated on HIV infected cells.
8. Result 2.6: How about the S1PR1 expression in the spleen? Is it downregulated in the spleen, so the cells are retaining in the spleen instead of egressing out of it? Since some of the humanized mouse models also developed lymph nodes, why aren’t the lymph nodes been examined?
9. Authors should include the in vitro migration assay and the use of S1PR1 modulators such as fingolimod in the study.
10. I suggest a cartoon illustration to demonstrate the interactions of CD69, S1PR1, TNF-a, INF-a and akt signaling in the thymus. HIV infected and non-infected cells should be included and show autocrine, paracrine or endocrine interactions. The S1P concentrations should also be described and included.
Author Response
Please see the attachment for the figure mentioned.
Reviewer 1
Resop et al. reported S1PR1 dysregulation during HIV-1 infection in human thymus using several humanized mouse models and in vitro cell culture experiments. They found S1PR1 expression is increased during HIV infection regardless of ART treatment, however, this effect can be inhibited by anti-IFNAR2. In vitro treatment of TNF-a and IFN-b can also increase S1PR1 expression in thymocytes. They also found that akt signaling is not impaired in the thymus during HIV infection. Since S1PR1 is upregulated in both CD69 positive and negative cells, the effect of CD69 on S1PR1 expression will need further investigation. Authors suggested that S1PR1 modulation may present a new therapeutic strategy for patients who lack immune system reconstitution due to the increased egress of immature thymocytes to the periphery.
Comments:
- Line 65: I am not an immunologist but my understanding of “mature” thymocytes means that they have become CD4 or CD8 single positive T cells. CD25 and CD69 are normally used as activation markers. Please explain why only CD8 is included in the flow panel, and any no CD8 nor CD4 results were shown to indicate the actual maturation of the thymocytes, especially in the humanized models when thymus epithelium cells are present.
We would first like to thank the reviewer for taking the time to read our manuscript and provide thoughtful feedback.
We agree with the reviewer that mature thymocytes are generally defined as those that are CD4 or CD8 single positive (SP). We included CD4, in addition to CD8, in all staining panels. We apologize for any omission of CD4 in the staining methods and have reviewed the manuscript to ensure that CD4 is now included in all mention of staining and in the methods. In our previous publication on S1PR in the thymus (Resop and Douaisi et al, Journal of Allergy and Clinical Immunology, 2016), we characterized the maturation stages of thymocytes using changing expression levels of CD3 and CD69 in addition to CD4 and CD8, as CD3/69 populations comprise both CD4 and CD8 double or single positive cells. The reviewer is correct that CD69 is generally utilized as an activation marker, which is in keeping with its expression throughout thymocyte development and selection and loss upon maturation and preparation for thymus egress (CD3hiCD69- mature thymocytes).
- Viral input (TCID) used for infecting mice should be included in the methods. Authors should also explain why 5 weeks, 9 weeks, and 12 weeks post infection were chosen since most of the background information is based on the findings in the acute infection phase.
We apologize for the omission and have updated the methods section with the relevant information. Specifically, for all humanized mouse experiments, animals were infected with 300-400 infectious units (IU) of virus.
The selection of time points was determined by the routes of humanized mouse infection (intrathymic or systemic).
For intrathymic infections where we directly infect the thymic implant, the 5- and 9-week time points were selected to safeguard against possible accelerated viral kinetics, which could result in severe cell depletion (typically, of CD3+69+ DP thymocytes) and impact results. For systemic infection, we selected the 12-week time point because, based on our prior studies, at this time we have detectable viral loads in peripheral blood, confirming established acute infection.
- Line 102: The reason for examining KLF2 should be explained in the introduction or result 2.1, not in the discussion.
We agree with the reviewer and have edited the manuscript to explain the rationale for examining KLF2 in the introduction and in detail in Results 2.1.
- Figure 1 (and most of the other data): authors should show how many percentages of these thymocytes were actually infected by HIV using flow. Flow cytometry gating strategies should include HIV p24 or other viral antigen to detect which cells were actually infected.
We appreciate this suggestion and completely agree with the reviewer that the inclusion of the percentage of thymocytes infected by flow cytometry would strengthen our studies. We also agree that ideally, quantitation of S1PR1 expression would include gating on infected cells and comparing S1PR1 in infected vs. non-infected cells. However, as the S1PR1 staining protocol involves a primary and secondary antibody, blocking, and then staining of all other surface markers, an intracellular p24 stain would be done following these steps. In our attempts to perform various versions of this staining procedure, this perturbed the quality of the S1PR1 stain. Although we have not yet been able to optimize this staining protocol, we are very open to suggestions or references to others who may have achieved intracellular and S1PR1 staining in combination, and we will bear this in mind for future work. Moreover, in future studies, if we obtain sufficient sample, it may be possible to sort infected and non-infected thymocytes and then perform RT-PCR for S1P receptor mRNA as an alternative/additional method. Unfortunately, we did not routinely obtain sufficient thymocytes from human thymic implants to perform separate staining for HIV viral antigens in addition to S1PR1.
- Figure 1 shows very high real time RT-PCR results presented as relative ratio compared to GAPDH. Some data points are close to 150-fold to GADPH. These are quite unusual real-time results. Or does GAPDH particularly low in the thymocytes?
We agree with the reviewer that for a few data points, especially for KLF2 in NL4-3 infected implants, there is unusually high expression of the target gene (KLF2) relative to GAPDH. We have reviewed the data to ensure that the relative expression values are accurate. For these notably high values, we hypothesize that the high KLF2 levels may be due to more robust infection in those human thymus implants and/or greater depletion of non-KLF2-expressing thymocytes. GAPDH is widely expressed in thymocytes, and we have used it routinely as a housekeeping gene for quantitative real-time RT PCR.
- Line 121 and 122: Is the concentration of IFN-a2 1000U or 10000U in mL? Are they physiologically relevant concentrations? Why isn’t the data included in the manuscript? Same to Line 126, is it U/ml?
We apologize that the concentration of IFN-a2 was not clearly explained. The concentrations used were 1,000 and 10,000 U/mL. We have updated the manuscript to state this clearly. We did not include the data in the manuscript because we were only able to perform these experiments on very few donors and, to the reviewer’s point, these concentrations may be too high to be physiologically relevant considering that the concentration in blood is much lower. However, as we used 2x10^7 thymocytes (a large number of cells) in 1mL of serum-free medium, we also used higher concentrations of cytokine. We agree with the reviewer that further experiments aimed at optimizing interferon concentrations are needed in more donors.
- Result 2.5: Why isn’t the phosphor-akt level increased by the increased of S1PR1? Therefore, there is no indication that the S1PR1 signaling is up-regulated by the increased expression of S1PR1. Cells should be gated on HIV infected cells.
We appreciate this point and the suggestions for future improvement of the phospho-Akt assay. We did see a small, non-statistically significant increase (trend toward increase) in pAkt signaling upon exposure to S1P in HIV-infected thymocytes, although this could arguably be due largely to a few data points with higher-than-average pAkt expression following S1P exposure. We agree that there is not strong support for increased S1P signaling with upregulated S1PR1 and we have therefore been careful to assert that there is simply no evidence of altered signaling (aka Akt signaling activity is maintained).
We also appreciate the suggestion to gate on HIV-infected cells and certainly would have preferred to do this. However, this would require either: 1. first sorting HIV-infected cells and then performing the pAkt assay, or 2. co-staining for viral antigens and pAkt, neither of which were possible. We did not generally obtain sufficient cells from implants to sort populations prior to downstream assays. Additionally, sorting definitively infected from non-infected cells would require permeabilization and detection of intracellular viral antigens, which would perturb the viability of cells and S1PR expression and function, precluding subsequent S1P exposure assays and pAkt staining. Further, as HIV antigen and pAkt staining also require different permeabilization reagents, co-staining p24 Gag and pAkt was not possible following the S1P exposure assay, so we were not able to gate on HIV-infected cells prior to analyzing pAkt, only total cells from infected vs. non-infected animals.
If there are other options that we may not be aware of, we are very open to suggestions and will use this information in future studies.
- Result 2.6: How about the S1PR1 expression in the spleen? Is it downregulated in the spleen, so the cells are retained in the spleen instead of egressing out of it? Since some of the humanized mouse models also developed lymph nodes, why aren’t the lymph nodes been examined?
We evaluated the expression of S1PR1 in the spleen (please see figure in word document of responses, submitted separately) and observed that splenic S1PR1 expression was somewhat elevated in HIV-infected mice, but this difference was not statistically significant across any of the populations examined. This suggests that retention/decreased egress from the spleen may not be likely, although further experiments designed to address this question would be necessary to answer this definitively.
It is true that some of the humanized mouse models develop lymph nodes. However, in the models used in our studies, peripheral lymph node development is not consistent, and when lymph nodes do develop, they yield a very small number of cells, making it highly challenging to obtain any meaningful results from the tissues.
Please see the uploaded word document for the figure.
- Authors should include the in vitro migration assay and the use of S1PR1 modulators such as fingolimod in the study.
Although outside of the scope of our current study, we have previously performed transwell migration assays using primary human thymocytes, fingolimod (FTY720), and other selective S1PR modulators. We published these experiments and results in Resop and Douaisi et al, Journal of Allergy and Clinical Immunology, 2016 (link below). In these experiments, we found that fingolimod-treated mature (CD3/27+) thymocytes had statistically significantly reduced migration to 10-100nM S1P relative to untreated.
We agree that transwell assays evaluating the migration potential of cells from human thymic implants in infected mice would be very interesting. Unfortunately, cells from implants in these animals are very limited and we generally did not have sufficient cells to perform all the experiments we wanted to do, such as migration assays. We appreciate the reviewer’s suggestion and will consider this for future studies.
Resop RS, Douaisi M, Craft J, Jachimowski LC, Blom B, Uittenbogaart CH. Sphingosine-1-phosphate/sphingosine-1-phosphate receptor 1 signaling is required for migration of naive human T cells from the thymus to the periphery. J Allergy Clin Immunol. 2016 Au
- I suggest a cartoon illustration to demonstrate the interactions of CD69, S1PR1, TNF-a, INF-a and Akt signaling in the thymus. HIV infected and non-infected cells should be included and show autocrine, paracrine or endocrine interactions. The S1P concentrations should also be described and included.
We would like to thank the reviewer for this suggestion and agree that a schematic illustration would be very helpful. We have now included a cartoon illustration demonstrating the interactions for CD69, S1PR1, TNF-a, IFN-a/b, IFNAR/aIFNAR blocking, and Akt in Supplemental Figure 4.
We have ensured that concentrations of S1P are clear in the manuscript.
We thank the reviewer for their time.

Reviewer 2 Report
In the current study, authors analyzed the consequences of HIV infection on S1P signaling and how the cytokines dysregulation would contribute to that. The manuscript is well-written and structured and I have only minor comments
1. S1PR1 signaling is associated with several physiological processes correlated to the immune system. The authors should mention that with some details in the introduction.
2. The quality of Figure 1 is low and should be improved.
3. In Figures 1B and C, the authors mentioned the increase of S1PR1 and KLF2 mRNA at week 5 in NL4-3 versus mock (while the P-value was not significant). It is clear that there was a trend at that time point that became significant at week 9 therefore I suggest rephrasing how the authors described that trend at week 5.
4. I wonder if the same signal was analyzed by other researchers under other viral infections and immune diseases. A comparative analysis in the discussion section would be interesting.
Minor editing of English language required
Author Response
Reviewer 2
In the current study, authors analyzed the consequences of HIV infection on S1P signaling and how the cytokines dysregulation would contribute to that. The manuscript is well-written and structured, and I have only minor comments
- S1PR1 signaling is associated with several physiological processes correlated to the immune system. The authors should mention that with some details in the introduction.
We would like to thank the reviewer for taking the time to provide constructive feedback on our paper. We agree that a discussion of the immune processes that S1P signaling is involved in is an important addition to the manuscript. We have modified the text of the introduction accordingly.
- The quality of Figure 1 is low and should be improved.
We apologize that the figure does not appear to be of good quality. We would appreciate information on the specific changes the reviewer would like us to make to the figure, and we would be happy to implement them. We have checked the quality and the figure is currently 25MB in size.
- In Figures 1B and C, the authors mentioned the increase of S1PR1 and KLF2 mRNA at week 5 in NL4-3 versus mock (while the P-value was not significant). It is clear that there was a trend at that time point that became significant at week 9 therefore I suggest rephrasing how the authors described that trend at week 5.
We agree with the reviewer. We have modified the wording of the weeks 5 and 9 results to reflect that there is a trend toward increased expression at week 5 and a statistically significant difference at week 9.
- I wonder if the same signal was analyzed by other researchers under other viral infections and immune diseases. A comparative analysis in the discussion section would be interesting.
We agree that it would be very interesting to include a comparative analysis in the discussion of S1P receptor dysregulation across various viral infections/immune conditions. To our knowledge, there is only one report that examined expression levels of S1PR1 in Newcastle disease virus infection. We have now included a brief paragraph in the discussion section.
We thank the reviewer for their time.
Reviewer 3 Report
This study by Resop et al. described how “HIV-1 infection results in Sphingosine-1-phosphate receptor 1 dysregulation in the human thymus”. It is a very well-designed and written study. My comments are below:
-In Figure 1. It would be interesting to compare the levels of normalized S1PR1 and KLF2 between weeks 5 and 9. Does the author have any plans to compare these levels between acute vs chronic HIV-1 infection in future work?
- Humanized Mice usage is a plus, nice work!
The paper can be accepted in its current form.
Author Response
Reviewer 3
This study by Resop et al. described how “HIV-1 infection results in Sphingosine-1-phosphate receptor 1 dysregulation in the human thymus”. It is a very well-designed and written study. My comments are below:
1. In Figure 1. It would be interesting to compare the levels of normalized S1PR1 and KLF2 between weeks 5 and 9. Does the author have any plans to compare these levels between acute vs chronic HIV-1 infection in future work?
We would like to thank the reviewer for taking the time to read our manuscript and provide constructive comments. We agree that it would be very interesting to compare levels of normalized S1PR1 and KLF2 in acute vs. chronic HIV-1 infection. As animals for these studies were limited, we selected two time points (weeks 5 and 9) to obtain information on the expression of S1PR1 and KLF2 at two times approximately representative of early and later acute infection.
We did not compare the average relative expression in the figure due to the limited number of observations—and not being able to compare the two time points within the same animal—but we agree that this would be a great avenue for future work. If we perform future studies examining peripheral T cells, for example, by collecting blood from the same animals at multiple time points, we would be able to perform comparisons within each animal as well as with average values.
A closer examination and comparison between acute and chronic infection would be a great avenue of investigation, especially if it is possible to leverage patient samples.
2. Humanized Mice usage is a plus, nice work!
3. The paper can be accepted in its current form.
We would like to thank the reviewer for taking the time to read our manuscript and provide their feedback.